# ESBL Activity, MDR, and Carbapenem Resistance among Predominant Enterobacterales Isolated in 2019

**DOI:** 10.3390/antibiotics10060744

**Published:** 2021-06-19

**Authors:** Altaf Bandy, Bilal Tantry

**Affiliations:** 1Family & Community Medicine, College of Medicine, Jouf University, 74311 Sakaka, Aljouf, Saudi Arabia; 2Ex-faculty, Department of microbiology, College of Medicine, Jouf University, 74311 Sakaka, Aljouf, Saudi Arabia; bilaltantry@gmail.com

**Keywords:** *Enterobacterales*, *E. coli*, *K. pneumoniae*, *P. mirabilis*, antimicrobials, multidrug resistance, carbapenem, ESBL

## Abstract

Antimicrobial-resistance in *Enterobacterales* is a serious concern in Saudi Arabia. The present study retrospectively analyzed the antibiograms of *Enterobacterales* identified from 1 January 2019 to 31 December 2019 from a referral hospital in the Aljouf region of Saudi Arabia. The revised document of the Centers for Disease Control (CDC) CR-2015 and Magiorakos et al.’s document were used to define carbapenem resistance and classify resistant bacteria, respectively. The association of carbapenem resistance, MDR, and ESBL with various sociodemographic characteristics was assessed by the chi-square test and odds ratios. In total, 617 *Enterobacterales* were identified. The predominant (*n* = 533 (86.4%)) isolates consisted of 232 (37.6%), 200 (32.4%), and 101 (16.4%) *Escherichia coli, Klebsiella pneumoniae,* and *Proteus mirabilis*, respectively. In general, 432 (81.0%) and 128 (24.0%) isolates were of MDR and ESBL, respectively. The MDR strains were recovered in higher frequency from intensive care units (OR = 3.24 (1.78–5.91); *p* < 0.01). *E. coli* and *K. pneumoniae* resistance rates to imipenem (2.55 (1.21–5.37); *p* < 0.01) and meropenem (2.18 (1.01–4.67); *p* < 0.04), respectively, were significantly higher in winter. The data emphasize that MDR isolates among *Enterobacterales* are highly prevalent. The studied *Enterobacterales* exhibited seasonal variation in antimicrobial resistance rates towards carbapenems and ESBL activity.

## 1. Introduction

Globally, infections with resistant Gram-negative bacteria are recognized as a severe threat to patients’ health. *Enterobacteriaceae* resistant to third-generation cephalosporins and carbapenems are critical microorganisms that require urgent attention [1]. A greater number of patients are at risk of dying from infections caused by carbapenem-resistant, extended-spectrum beta-lactamase (ESBL)-producing, and multidrug-resistant (MDR) *Enterobacteriacea* [2]. Carbapenemase-producing *Enterobacteriaceae* infections are associated with a very high case fatality rate with currently available antibiotics [3,4]. Intrinsic and acquired bacterial resistance, international travel, inappropriate antibiotic use, and poor infection control activities are predisposing factors for developing resistance [5,6].

Global dissemination of resistant *Enterobacteriaceae* is increasing [7,8]. The Kingdom of Saudi Arabia receives numerous pilgrims throughout the year, particularly in the holy month of Hajj, raising the possibility of resistant *Enterobacteriaceae* transmission. International travel is a serious concern for the transmission of resistant *Enterobacteriaceae* in the country [9]. Various studies and reports emerging from different regions of the kingdom have highlighted the spread of such resistant strains [10,11,12,13,14]. Furthermore, the expatriate population in the kingdom is a matter of concern considering the transmission of *Enterobacteriaceae* with diverse resistance mechanisms [15].

The surveillance of resistant strains is a prerequisite for control measures [16]. Insights into antibiotic resistance provide valuable information about significant pathogens and guide actions of antimicrobial stewardship programs, infection control committees, and public health agencies [17]. No exclusive study has reported antimicrobial resistance in Enterobacterales from the Aljouf region. The present study was conducted to determine carbapenem resistance, ESBL activity, and multidrug resistance among predominant Enterobacterales from a referral hospital in the Aljouf region of Saudi Arabia.

## 2. Results

### 2.1. Enterobacterales and Associated Sociodemographic Characteristics

Of the 617 Enterobacterale isolates, 232 (37.6%), 200 (32.4%), and 101 (16.4%) isolates were of *E. coli, K. pneumoniae*, and *P. mirabilis*, respectively. The other less frequent Enterobacterales included 27 (4.4%) *Enterobacter aerogenes* (*Klebsiella aerogenes*), 22 (3.6%) *Enterobacter cloacae* (*Klebsiella cloacae*), 15 (2.4%) *Providencia stuartii,* 13 (2.1%) *Serratia marcescens*, and 7 (1.1%) *Morganella morganii.* Further analysis of the 533 predominant *Enterobacterales* revealed that various sociodemographic characteristics such as sex, specimen type, referring unit, and resistance type were significantly (*p* < 0.05) associated with the distribution of *E. coli, K. pneumoniae*, and *P. mirabilis*. In general, 432 (81.0%), 97 (18.2%), and 15 (2.8%) isolates were MDR, XDR, and PDR, respectively. Of the 432 multidrug-resistant isolates, 167 (38.6%), 170 (39.3%), and 95 (21.9%) isolates were *E. coli*, *K. pneumoniae*, and *P. mirabilis*, respectively. Furthermore, 128 (24.0%) of these *Enterobacterales* were extended-spectrum beta-lactamase (ESBL) producers (Table 1).

### 2.2. MDR, ESBL, and Associated Sociodemographic Characteristics

The multidrug-resistant organisms were isolated with significantly higher frequency from clinical samples of intensive care units (OR = 3.24 (1.78–5.91); *p* < 0.01) compared to the samples received from surgical wards and other hospitals (OR = 2.67 (1.29–5.50); *p* < 0.01). Other sociodemographic characteristics did not reveal any significant association with multidrug strains (Table 2).

The ESBL isolates were observed with significantly (*p* < 0.05) lesser frequency in the samples received in the second, third, and fourth quarters of the year compared to the first quarter (January–March) (Table 3).

### 2.3. Carbapenem Resistance and Associated Sociodemographic Characteristics

*K. pneumoniae* exhibited 62.5%, 62%, and 58.3% resistance towards ertapenem, imipenem, and meropenem, respectively. A resistance rate of 26.5% towards meropenem was observed in *P. mirabilis.*

Meropenem resistance in *P. mirabilis* was significantly higher among the male patients (5.5 (1.72–17.56); *p* < 0.01) and patients aged ≥50 years (4.15 (1.29–13.30); *p* < 0.02). In *E. coli* and *K. pneumoniae*, seasonal variation in the antimicrobial resistance rate was observed for imipenem (2.55 (1.21–5.37); *p* < 0.01) and meropenem (2.18 (1.01–4.67); *p* < 0.04), respectively. Furthermore, the *K. pneumoniae* meropenem resistance rate was significantly higher in samples received from intensive care units than from other units (1.92 (1.08–3.43); *p* < 0.02) (Table 4).

### 2.4. Antimicrobial Resistance among Predominant Enterobacterales

Aminoglycosides (amikacin and gentamycin) are effective antibiotics against *E. coli* and *K. pneumoniae*. *E. coli* and *K. pneumoniae* exhibited sensitivities of 96.5% and 74.9% towards amikacin, respectively. First-generation cephalosporin (cephalothin) was ineffective, and a resistance rate of >80% was observed. All the studied bacteria presented >50% resistance to second- through fourth-generation cephalosporins, except for cefoxitin. Third-generation cephalosporins were less effective against *K. pneumoniae* (27.4% highest for ceftriaxone) and *P. mirabilis* (25.0% highest for ceftazidime). *P. mirabilis* exhibited highest resistance (76.8%) towards cefepime (fourth-generation cephalosporin) (Table 5).

The meropenem multidrug-resistant (carbapenem-resistant *Enterobacteriaceae*) strains of *P. mirabilis* and *E. coli* exhibited the highest (100.0%) and lowest (47.0%) resistance rates to tigecycline, respectively. Almost all (96.0%) of the tested meropenem multidrug-resistant strains of *P. mirabilis* were resistant to colistin (Figure 1).

## 3. Discussion

Antimicrobial resistance is prevalent worldwide, and the mortality due to drug-resistant infection is predicted to reach 10 million annually by 2050 [18]. The current treatment options for drug-resistant infections are becoming less effective, thereby leading to treatment failures, increasing treatment cost, and endangering life [1]. Antimicrobial resistance, particularly carbapenem resistance in Gram-negative bacteria, is the latest challenge of immense public health importance [19]. Carbapenemase-producing *Enterobacteriaceae* such as KPC and New Delhi metallo-β-lactamase NDM have achieved global dissemination and pose a serious threat to human life [8,20]. The Kingdom of Saudi Arabia has reported a rise in carbapenem-resistant *Enterobacteriaceae*, mostly from Riyadh [15,21,22,23,24] and Makkah [14,25].

In the present study, *K. pneumoniae, E. coli, and P. mirabilis* were the major *Enterobacterales* isolated during the study period. These findings are in accordance with earlier studies from Saudi hospitals and countries in the MENA region [20,26,27]. These microorganisms have achieved remarkable antimicrobial resistance and have been increasingly recovered from various clinical samples [27].

In general, a very high prevalence (82.1%) of MDR was observed in this study. The majority of MDR isolates belonged to *K. pneumoniae* and *E. coli* in almost equal proportions. Furthermore, this study reported 128 (24.0%) isolates as extended-spectrum beta-lactamase (ESBL) producers. *E. coli* was the most common ESBL active, followed by *K. pneumoniae*. The antimicrobial resistance and prevalence of MDR or ESBL activity among bacteria are difficult to compare and depend on various local factors such as region- and country-specific guidelines on antimicrobial use, self-medication, infection control practices, and practices affecting antimicrobial prescription and use [28]. Furthermore, study-specific factors such as analysis of clinical samples from inpatients, outpatients, or both; type of clinical sample analyzed (urine, blood, sputum, or all types of samples); treating intermediate resistance as resistant or sensitive categories; and the analysis adopted to measure various rates affect comparison across studies. A recent study evaluating the effectiveness of the Saudi Action Plan in combating antimicrobial resistance identified *E. coli* as the most frequently isolated *Enterobacterale* and the predominant bacteria for ESBL activity and MDR prevalence, followed by *K. pneumoniae* [29]. Similarly, another study from the neighboring country of Sudan reported *E. coli* and *K. pneumoniae* as the most commonly isolated *Enterobacterales*. *E. coli* was reported as a frequent MDR isolate, whereas ESBL activity was more frequent in *K. pneumoniae* [27]. These reports are in accordance with our results on the most common bacterial isolation and ESBL activity, with a slight variation in MDR prevalence among the two bacteria. A study from Addis Ababa reported a 42.1% MDR among *Enterobacteriaceae* [30]; however, another recent study from Saudi Arabia reported an overall prevalence of 60% MDR *Enterobacteriaceae*, with *E. coli* and *K. pneumoniae* as the most frequent isolates [31], but this prevalence was only 6.6% in the USA hospitals [32]. The analysis adopted in the present study revealed higher MDR prevalence, and this prevalence was determined for the three predominant bacteria and not for all *Enterobacterales* isolates reported during the study period. The extended-spectrum penicillin and cephalosporins are safer than other antimicrobials, resulting in their wide use in clinical settings [33], which might partly explain the increased MDR prevalence and ESBL activity in these bacteria. Of the 101 isolates of *P. mirabilis* recovered, 95 (94%) were MDR. Our results on the MDR prevalence are consistent with a study from India that reported a >95% MDR rate in *P. mirabilis* [34]. The biofilms produced by this organism not only help to evade the lethal effect of antimicrobials but also induce bacterial changes to resist and escape host defenses [35,36]. The free availability of antimicrobials and non-adherence, which are frequent in Saudi Arabia, could be attributed to this increased resistance and subsequent MDR prevalence.

The MDR isolates were recovered from intensive care units at a significantly higher frequency, whereas ESBL isolates were significantly higher in winter. Other studies have reported similar results regarding the frequency of MDR in intensive care units [32,37]. The reasons for this observed phenomenon are that patients admitted to intensive care units have multiple pathologies and are frequently treated with empirical antibiotic therapy before culture and sensitivity reports compared to non-intensive care unit patients. The high frequency of ESBL isolation in winter may be a result of higher antimicrobial consumption during these months. Seasonal variation in antimicrobial resistance has been previously reported [38].

In the present study, *K. pneumoniae* isolates exhibited resistance rates of 62.5%, 62% and 58% for ertapenem, imipenem, and meropenem, respectively. *E. coli* isolates revealed an overall carbapenem resistance of 10.9%, with the highest rate of 15.8% for imipenem. Moreover, 29.3% and 25.8% resistance to ertapenem and meropenem was observed in *P. mirabilis*, respectively. Ali Al Bshabshe et al. reported 65.2% and 61.7% resistance in *K. pneumoniae* towards ertapenem and meropenem, respectively, which is consistent with our results [39]. The slight increase in these resistance rates is because their study analyzed clinical samples from intensive care units only. In another study, 38.4% and 46.1% resistance rates to imipenem and meropenem, respectively, were noted in *K. pneumoniae* [40]. Carbapenem resistance is a concern as these drugs are the treatment of choice, particularly against ESBL-producing *K. pneumoniae*. These rates indicate the emergence of carbapenemase-producing strains. Timely identification of carbapenem strains is pivotal for controlling these infections in hospital settings. A previous study reported that 43.5% of antimicrobials were inappropriately prescribed; of these, broad-spectrum activity (44.6%) and use of antimicrobials without culture (32.4%) were the main reasons [41]. Inappropriate prescriptions when coupled with self-medication and improper antibiotic consumption may greatly enhance antimicrobial resistance. Appropriate antibiotic prescription guidelines, regulation of antibiotic dispensing by community pharmacies, and patient education are some of the vital measures in combating the menace of antimicrobial resistance.

In the present study, *E. coli* and *K. pneumoniae* presented a significant seasonal variation (winter vs. other seasons) in the resistance rates for imipenem (2.55 (1.21–5.37); *p* < 0.01) and meropenem (2.18 (1.01–4.67); *p* < 0.04), respectively. A study from the USA covering hospitalized patients did not find significant seasonal variation in carbapenem resistance among *Enterobacteriaceae* [42]. Although this phenomenon needs further exploration, this variation can be explained from the fact that both antibiotic consumption and antimicrobial resistance increases in the winter season [38]. Furthermore, the present study observed approximately 30% of the *Enterobacteriaceae* infections in winter, which may explain the different resistance rates towards carbapenems.

Colistin and tigecycline are used against carbapenem-resistant *Enterobacteriaceae* (CRE). This study evaluated the antimicrobial activity of these antibiotics against carbapenem-resistant (meropenem-resistant) and multidrug-resistant strains identified in this study. Tigecycline exhibited 53% and 45% sensitivity against *E. coli* (CRE) and *K. pneumoniae* (CRE), respectively. Similarly, colistin was 100% and 82.1% effective against these two strains, respectively. Both antibiotics were ineffective against *P. mirabilis.* The treatment options for carbapenem-resistant strains are limited, that is, the current standard treatment for such infections includes tigecycline, polymyxins, fosfomycin, and aminoglycosides, which are administered alone or in combination with other antibiotics, the latter revealing better outcomes. An increase in tigecycline sensitivity from 76% to 82% against *E. coli* (CRE) and from 33% to 50% for *K. pneumoniae* (CRE) has been reported in Riyadh. The same study reported a decline in colistin sensitivity from 97% to 86% against *E. coli* (CRE) and from 80% to 76% against *K. pneumoniae* (CRE) [43]. The observations from Riyadh complement our findings on these two antibiotics, with subtle variations. Biofilms and intrinsic resistance to colistin and tigecycline are prominent features of *P. mirabilis,* which explains its resistance towards these antibiotics [44].

The major strength of this study is that it is the first exclusive report on the resistance pattern in *Enterobacterales* from this region. Furthermore, the study analyzed the recent data of 2019, which is the second strength of the study. One limitation of this study is that it included data from only one referral center out of two operating in the capital city of the Aljouf region, thereby limiting the generalizability of the results; however, the study hospital is the oldest in the region and continues to process the majority of clinical samples in addition to those received from other hospitals. Another limitation is the lack of molecular analysis of this acquired resistance.

## 4. Conclusions

The study concludes that a high prevalence of MDR and carbapenem resistance exist in the study area. Intensive care units majorly contribute to the prevalence of MDR. Seasonal variations in ESBL-producing bacteria and carbapenem resistance were noted. High tigecycline resistance was noted, which has an immense impact on treatment options and patient outcomes. Antimicrobial resistance rates are worrying and require immediate action at the institutional and public health levels. Strict infection control practices and antimicrobial prescription guidelines need to be observed in the intensive care units to counter antimicrobial resistance. The seasonal variation in carbapenem resistance should guide empirical antibiotic therapy in this hospital. Identification of the molecular basis of the observed resistance and exploring the reasons for the seasonal variation are potential areas for future research.

## 5. Material and Methods

The Aljouf region has three governorates of Sakaka, Qurayyat, and Dumat Al-Jandal. The total population of the Aljouf region is 520,737 as per the 2018 census. The Sakaka governorate is the capital city of the Aljouf region with two specialist hospitals of 300 beds each. The study was conducted in one of the two specialist hospitals that serve the population of this region. This hospital is the first specialist hospital built in the region. This retrospective hospital-based study analyzed the data on culture and antibacterial reports of all non-duplicate *Enterobacteriaceae* isolated from 1 January 2019 to 31 December 2019. The BD Phoenix system (BD Diagnostics, Sparks, MD, USA) processed all the samples and generated a sensitivity report.

The antibiotics used to assess bacterial susceptibility were imipenem, meropenem, ertapenem, amikacin, gentamycin, aztreonam, ampicillin, augmentin, piperacillin/tazobactam, cephalothin, cefoxitin, cefuroxime, ceftazidime, ceftriaxone, cefepime, nitrofurantoin, ciprofloxacin, levofloxacin, tigecycline, colistin, and trimethoprim/sulfamethoxazole. Intermediate resistance presented by the bacterial isolates was interpreted as being resistant. Magiorakos et al.’s criteria were used for classification of antimicrobial resistance into multidrug resistance (MDR), extensive drug resistance (XDR), and pan drug resistance (PDR) [45]. Multidrug resistance (MDR) is defined as when a bacterium is ‘non-susceptible to ≥1 agent in ≥3 antimicrobial categories’. Extensive drug resistance (XDR) is defined as when a bacterium is ‘non-susceptible to ≥1 agent in all but ≤2 categories (i.e., bacterial isolates remain susceptible to only one or two categories), while pan drug resistance (PDR) is defined as when a bacterium is ‘non-susceptible to all antimicrobial agents listed’ [45]. Moreover, carbapenem-resistant *Enterobacteriaceae* (ER) was defined according to the revised document of the Centers for Disease Control and Prevention CR-2015 [46]. The bacterial antimicrobial susceptibility results were interpreted using breakpoints established by the Clinical Laboratory Standards Institute (CLSI) [47]. Information on the phenotypic characterization of ESBL provided by the BD Phoenix system was also recorded. Furthermore, the study collected data on the type of specimen, specimen collection date, information of different units referring the sample, and age and gender of patients.

### Statistics

Statistical Package for Social Sciences (SPSS) version 21.0 for Windows (SPSS, Inc., Chicago, IL, USA) was used for data analysis. The entries were manually verified for accuracy. The frequencies and percentages of *Enterobacteriaceae* distribution and MDR isolates were calculated. Additionally, the association between sociodemographic and distribution characteristics of *Enterobacteriaceae* was assessed using the chi-square test with or without Yates correction. Fisher’s exact test was performed whenever necessary. Furthermore, the distribution characteristics of multidrug-resistant strains and various carbapenem-resistant isolates vis-à-vis sociodemographic risk factors such as sex, age groups, seasonality, referring unit, and type of sample were determined by odds ratios with 95% confidence intervals. Statistical significance was set at *p* < 0.05.

## Figures and Tables

**Figure 1 antibiotics-10-00744-f001:**
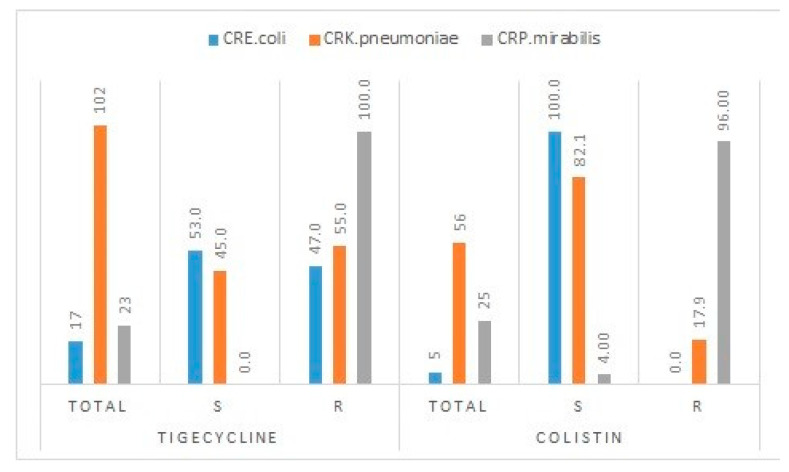
Tigecycline and colistin resistance in carbapenem-resistant *E*. *coli*, *K*. *pneumoniae,* and *P*. *mirabilis* (total in absolute numbers, and individual resistance in percentages).

**Table 1 antibiotics-10-00744-t001:** Distribution characteristics and resistance profile in *E*. *coli*, *K*. *pneumoniae*, and *P*. *mirabilis* (*n* = 533).

Characteristic		*E*. *coli* (232)	*K*. *pneumoniae* (200)	*P*. *mirabilis* (101)	*p* Value
Sex	Male	89	118	59	0.01
	Female	143	82	42	
Age group (years)	≥60	100	97	51	0.05
	40–59	39	51	21	
	20–39	68	37	20	
	≤19	25	15	9	
Seasonality	Quarter 1	75	85	37	0.37
	Quarter 2	55	47	21	
	Quarter 3	44	28	20	
	Quarter 4	58	40	23	
* Specimen type	Urine	161	97	38	0.01
	Blood	23	63	24	
	Wound swab	48	40	8	
** Referring unit	Intensive care units	63	120	61	0.01
	Outpatient departments	64	29	21	
	Surgical wards	45	27	9	
	Medical wards	24	10	10	
*** Resistance Type	MDR	167	170	95	0.01
	XDR	8	65	24	
	PDR	2	12	1	
	ESBL	85	26	17	

* From tracheal wash (8) and sputum (23) samples *P*. *mirabilis* was the only isolate, not included in analysis. ** *P. mirabilis* was not isolated from specimens referred from other hospitals, not included in analysis. *** Yates correction.

**Table 2 antibiotics-10-00744-t002:** Sociodemographic distribution characteristics of multidrug-resistant isolates (*n* = 432).

Characteristic		MDR	Non–MDR	OR (95% CI)	*p* Value
Sex	Male	213	53	0.88 (0.57–1.35)	0.64
	Female	219	48		
Age group (years)	≥60	204	44	1	
	40–59	89	22	1.14 (0.65–2.02)	0.64
	20–39	96	29	1.40 (0.82–2.37)	0.21
	≤19	43	6	0.64 (0.25–1.61)	0.46
Seasonality	Quarter 1	155	42	1	
	Quarter 2	101	22	0.80 (0.45–1.42)	0.54
	Quarter 3	72	20	1.02 (0.56–1.87)	0.92
	Quarter 4	104	17	0.60 (0.32–1.11)	0.14
* Specimen type	Urine	238	58	1	
	Blood	82	12	0.60 (0.30–1.17)	0.17
	Wound swab	81	31	1.57 (0.94–2.59)	0.10
Referring unit	Intensive care units	213	31	1	
	Outpatient departments	92	22	1.64 (0.90–2.98)	0.10
	Surgical wards	55	26	3.24 (1.78–5.91)	0.01
	Medical wards	36	8	1.52 (0.65–3.58)	0.46
	Referred from other hospitals	36	14	2.67 (1.29–5.50)	0.01

* *P*. *mirabilis* was the only *Enterobacterale* isolated from tracheal wash (8) and sputum (23) samples, not included in analysis.

**Table 3 antibiotics-10-00744-t003:** Sociodemographic distribution characteristics of ESBL isolates (*n* = 128).

Characteristic		ESBL	Non–ESBL	OR (95% CI)	*p* Value
Sex	Male	61	205	0.88 (0.59–1.32)	0.55
	Female	67	200		
Age group (years)	≥60	67	181	1	
	40–59	21	90	1.58 (0.91–2.75)	0.09
	20–39	28	97	1.28 (0.77–2.15)	0.19
	≤ 19	12	37	1.14 (0.56–2.31)	0.71
Seasonality	Quarter 1	32	165	1	
	Quarter 2	35	88	0.48 (0.28–0.84)	0.01
	Quarter 3	25	67	0.51 (0.28–0.94)	0.02
	Quarter 4	36	85	0.45 (0.26–0.78)	0.01
* Specimen type	Urine	70	226	1	
	Blood	26	68	0.81 (0.47–1.37)	0.43
	Wound swab	23	89	1.19 (0.70–2.03)	0.50
Referring unit	Intensive care units	49	195	1	
	Outpatient departments	24	90	0.94 (0.54–1.63)	0.82
	Surgical wards	21	60	0.71 (0.39–1.29)	0.26
	Medical wards	18	26	0.36 (0.18–0.71)	0.01
	Referred from other hospitals	16	34	0.53 (0.27–1.04)	0.06

* *P*. *mirabilis* was the only *Enterobacterale* isolated from tracheal wash (5) and sputum (4) samples, not included in analysis.

**Table 4 antibiotics-10-00744-t004:** Carbapenem resistance distribution characteristics of predominant *Enterobacterales*.

Organism	Characteristic		Ertapenem		Odds Ratio (95% CI)	*p* Value	Imipenem		Odds Ratio (95% CI)	*p* Value	Meropenem		Odds Ratio (95% CI)	*p* Value
			R	S			R	S			R	S		
*E*. *coli*	Sex	Male	7	81	1.03 (0.38–2.78)	0.92	11	76	0.67 (0.31–1.44)	0.30	8	79	0.99 (0.39–2.5)	1.0
		Female	11	132			25	116			13	128		
*K*. *pneumoniae*		Male	72	46	0.85 (0.47–1.53)	0.60	69	49	0.69 (0.38–1.24)	0.21	65	52	0.75 (0.42–1.35)	0.35
		Female	53	29			55	27			51	31		
** *P. mirabilis*		Male	18	40	1.22 (0.50–2.97)	0.64	–	–	–	–	22	36	5.5 (1.72–17.56)	0.01*
		Female	11	30			–	–			4	36		
*E*. *coli*	Age group	>50	9	108	0.97 (0.37–2.54)	1.0	17	100	0.82 (0.40–1.67)	0.59	12	105	1.29 (0.52–3.20)	0.69
		≤ 50	9	105			19	92			9	102		
*K*. *pneumoniae*		>50	67	48	0.64 (0.36–1.17)	0.15	69	46	0.81 (0.45–1.46)	0.49	64	50	0.81 (0.45–1.43)	0.27
		≤50	58	27			55	30			52	33		
** *P. mirabilis*		>50	22	42	2.09 (0.78–5.55)	0.13	–	–	–	–	22	41	4.15 (1.29–13.30)	0.02*
		≤50	7	28			–	–			4	31		
*E*. *coli*	Seasonality	Quarter 4	8	50	2.60 (0.97–6.96)	0.84	15	42	2.55 (1.21–5.37)	0.01	8	49	1.98 (0.77–5.06)	0.14
		Quarters 1, 2 and 3	10	163			21	150			13	158		
*K*. *pneumoniae*		Quarter 4	30	10	2.06 (0.93–4.48)	0.06	30	10	2.10 (0.96–4.60)	0.05	29	11	2.18 (1.01–4.67)	0.04
		Quarters 1, 2 and 3	95	65			94	66			87	72		
** *P. mirabilis*		Quarter 4	4	19	0.42 (0.13–1.39)	0.24 *	–	–	–	–	6	16	1.05 (0.36–3.05)	0.92
		Quarters 1, 2 and 3	25	51			–	–			20	56		
*E*. *coli*	Specimen type	Blood	1	22	0.51 (0.06–4.02)	0.70	1	22	0.22 (0.02–1.69)	0.13	0	23	–	–
		Others	17	191			35	170			21	184		
*K*. *pneumoniae*		Blood	39	24	0.96 (0.52–1.78)	0.92	41	22	1.21 (0.65–2.25)	0.54	40	23	1.37 (0.74–2.53)	0.31
		Others	86	51			83	54			76	60		
** *P. mirabilis*		Blood	1	7	0.32 (0.03–2.73)	0.43	–	–	–	–	0	7	–	–
		Others	28	63			–	–			26	65		
*E*. *coli*	Specimen referring unit	ICU	8	54	2.35 (0.88–6.27)	0.09	10	52	1.03 (0.46–2.29)	0.92	8	55	1.70 (0.66–4.32)	0.25
		Others	10	159			26	140			13	152		
*K*. *pneumoniae*		ICU	79	41	1.42 (0.79–2.54)	0.23	81	39	1.78 (0.99–3.19)	0.05	77	42	1.92 (1.08–3.43)	0.02
		Others	46	34			43	37			39	41		
** *P. mirabilis*		ICU	15	45	0.59 (0.24–1.43)	0.24	–	–	–	–	13	45	0.60 (0.24–1.48)	0.26
		Others	14	25			–	–			13	27		

* Yates correction; ** *P. mirabilis* shows higher minimum inhibitory concentration for imipenem, hence resistance towards ertapenem, and meropenem depicts carbapenem resistance; R = Resistant; S = Sensitive

**Table 5 antibiotics-10-00744-t005:** Antimicrobial sensitivity of the studied organisms.

Antibiotic	*E*. *coli* (%)	*K*. *pneumoniae* (%)	*P*. *mirabilis* (%)
Amikacin	96.5	74.9	51.4
Gentamycin	79.3	63.8	10.9
Cephalothin	11.8	12.7	12.5
Cefuroxime	41.1	17.5	20.0
Cefoxitin	81.8	38.0	78.2
Ceftazidime	48.0	26.0	25.0
Ceftriaxone	45.7	27.4	21.8
Cefepime	47.8	31.3	23.2
Aztreonam	47.6	25.0	38.1
Ampicillin	15.6	0.5	14.0
Amoxcyline–clavunate	34.1	19.2	19.0
Piptazobactam	82.7	33.5	76.0
Trimethoprim–Sulfamethoxazole	38.3	33.7	15.8
Nitrofurantoin	82.5	22.4	0.0
Ciprofloxacin	40.8	31.0	13.1
Levofloxacin	45.0	35.9	13.1

## Data Availability

The datasets generated and/or analyzed during the current study are available from the corresponding author on reasonable request. The dataset will be uploaded in the relevant repository after acceptance.

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
