# Peer review of "ESBL Activity, MDR, and Carbapenem Resistance among Predominant Enterobacterales Isolated in 2019"

_antibiotics, 2021, doi:10.3390/antibiotics10060744_

Round 1

Reviewer 1 Report

The study aimed to determine carbapenem-resistance, ESBL activity and multidrug-resistance among Enterobacterales (K. pneumonaie, P. mirabilis, E.coli) from a referral hospital in the Aljouf region in Saudi Arabia. Although the study can add relevant findings helpful to better describe epidemiology of multidrug resistant microorganisms, several revisions are required.

Enterobacteriaceae resistant to 3rd third-generation cephalosporins and carbapenems are a critical list of microorganisms that require urgent attention add reference i.e.  doi: 10.1111/lam.13384.

Insights 47 into antibiotic resistance provide valuable information about significant pathogens and 48 guide actions of antimicrobial stewardship programs, infection control committees, and 49 public health agencies. add reference i.e.  doi: 10.2147/IDR.S226416.

MDR, XDR and PDR, respectively suggest to add explanation and interpretation of these categories

The meropenem multidrug-resistant (CRE) strains of P. mirabilis CRE Should be explained in full terminology

The antimicrobial resistance and prevalence of MDR or ESBL activity among bacteria are difficult to compare and depend on various local factors such as region-and country-specific guidelines on antimicrobial use, self-medication, infection control practices and practices affecting antimicrobial prescription and use. Furthermore, study-specific factors such as analysis of clinical samples from inpatients, outpatients or both; type of clinical sample analysed (urine, blood, sputum, or all types of samples); treating intermediate resistance as resistance or sensitive categories and the analysis adopted to measure various rates. Could you add some reference?

The antibiotics used to assess bacterial susceptibility explain the method to assess antimicrobial susceptibility and how was carried out interpretation of the results

Table 1. the same classification level for each of the considered variable should be used  as in the table 4 for better comparison of the results

Table 2. should be avoided and add only some results in the text

Table 4. it should be simplified for better comprehension, moreover as odds ratio is included should be also mentioned methods for achieving these results in the methods section

Figure 1. is unnecessary, as results can be included in the text and already in the table 4

ESBL activity should be more explained in terms of the results

Author Response

Dear reviewer

Thanks for your suggestions that have helped to better shape the manuscript. The authors have incorporated the suggestions.

Thanks and regards 

Response to reviewer-1

Queries addressed

Q-1: Enterobacteriaceae resistant to 3rd third-generation cephalosporins and carbapenems are a critical list of microorganisms that require urgent attention add reference i.e.  doi

Response: doi added.

Q-2: Insights 47 into antibiotic resistance provide valuable information about significant pathogens and 48 guide actions of antimicrobial stewardship programs, infection control committees, and 49 public health agencies. add reference i.e.  Doi: 10.2147/IDR.S226416.

Response: Added

Q-3: MDR, XDR and PDR, respectively suggest to add explanation and interpretation of these categories.

Response: Explanation added in methods section.

Q-4: The meropenem multidrug-resistant (CRE) strains of P. mirabilis CRE Should be explained in full terminology

Response: Explained in full terminology.

Q-5: The antimicrobial resistance and prevalence of MDR or ESBL activity among bacteria are difficult to compare and depend on various local factors such as region-and country-specific guidelines on antimicrobial use, self-medication, infection control practices and practices affecting antimicrobial prescription and use. Furthermore, study-specific factors such as analysis of clinical samples from inpatients, outpatients or both; type of clinical sample analysed (urine, blood, sputum, or all types of samples); treating intermediate resistance as resistance or sensitive categories and the analysis adopted to measure various rates. Could you add some reference?

Response: Reference added in the discussion section.

Q-6: The antibiotics used to assess bacterial susceptibility explain the method to assess antimicrobial susceptibility and how was carried out interpretation of the results

Response: Explained in methods section with reference.

Q-7: Table 1. The same classification level for each of the considered variable should be used as in the table 4 for better comparison of the results

Response: The same classification level for each variable has been considered expect that it presents them vis-à-vis with each microorganism.

Q-8: Table 2. Should be avoided and add only some results in the text.

Response: Thanks for the suggestion, I wish to keep it. Please note it follows the objectives of the study exactly. Only the important findings are highlighted in results section.

Q-9: Table 4. it should be simplified for better comprehension, moreover as odds ratio is included should be also mentioned methods for achieving these results in the methods section.

Response: Odds ratio has been mentioned in the methods section and table 4 is self-explanatory.

Q-10: Figure 1. is unnecessary, as results can be included in the text and already in the table 4

Response: Thanks. Figure-1removed.

Q-11: ESBL activity should be more explained in terms of the results.

Response: It has been discussed and explained in discussion section. In the results sections we have highlighted the findings only.

Reviewer 2 Report

Manuscript sumarizes characterization of different antibiotic resistant isolates from hospital in view of MDR, ESBL and carbapenem resistance. Authors should decribe more about the situation in hospitals in introduction. Overall, writing of the latin names of bacteria should be improved, not everithing is in italic. Also authors should write antibiotic resistance or antimicrobial resistance, not only resistance or resistant because there are many factors/compounds to which bacteria can display the resistance.

I think that section materials and methods should be improved significantly. There shoud be better description of hospitals from which comes the isolates. I miss the identification method of bacteria. Also antimicrobial testing description is missing, only applied antibiotics are mentioned. No information about applied concentration, if CLSI or EUCAST was applied, no information about methodoogy. In this form, it is very insufficient. Did the authors used any control strain during testing?

Author Response

Dear reviewer

Thanks for your suggestions.

The authors have incorporated your suggestions that have benefitted the manuscript immensely.

Thanks and regards

Response to reviewer-2

Queries addressed

Q-1: Manuscript sumarizes characterization of different antibiotic resistant isolates from hospital in view of MDR, ESBL and carbapenem resistance. Authors should decribe more about the situation in hospitals in introduction. Overall, writing of the latin names of bacteria should be improved, not everithing is in italic. Also authors should write antibiotic resistance or antimicrobial resistance, not only resistance or resistant because there are many factors/compounds to which bacteria can display the resistance.

Response: Addressed where ever necessary.

Q-2: I think that section materials and methods should be improved significantly. There shoud be better description of hospitals from which comes the isolates. I miss the identification method of bacteria. Also antimicrobial testing description is missing, only applied antibiotics are mentioned. No information about applied concentration, if CLSI or EUCAST was applied, no information about methodoogy. In this form, it is very insufficient. Did the authors used any control strain during testing?

Response: Thanks addressed the issues in the methods section.

Round 2

Reviewer 1 Report

The Authors addressed the revisions required and manuscript improved. 

Author Response

Dear Sir

Thank you very much for your suggestions.

A spelling check has been carried out and necessary corrections have been made.

Thanks and regards

Reviewer 2 Report

Authors changed only one word into intalic, but there are more words that should be written in italic (for example Enterobacteriaceae). Please check the manuscript and correct every word. In my review I also encouraged authors to improve description of hospital. I still miss some characteristics such number of beds, type of treatment of hospital effluent etc. According to this, I think, that manuscript still could be improved.

Author Response

Dear Sir 

Thank you very much.

The words that need to be changed to italic font has been done.

Hospital description added.

Thanks and regards